# Direct evaluation of self-quenching behavior of fluorophores at high concentrations using an evanescent field

**Wooli Bae**[1]*, **Tae-Young Yoon**[2,3], **Cherlhyun Jeong**[4,5]*

**1** Imperial College Centre for Synthetic Biology and Department of Bioengineering, Imperial College London, South Kensington Campus, London, United Kingdom, **2** School of Biological Sciences, Seoul National University, Seoul, South Korea, **3** Institute for Molecular Biology and Genetics, Seoul National University, Seoul, South Korea, **4** Center for Theragnosis, Korea Institute of Science and Technology, Seoul, Republic of Korea, **5** KHU-KIST Department of Converging Science and Technology, Kyunghee University, Seoul, Republic of Korea

\* w.bae@imperial.ac.uk (WB); che.jeong@kist.re.kr (CJ)

**Data Availability Statement:** The raw data is deposited in Zenodo with DOI 10.5281/zenodo. 4525017.

**Funding:** W.B. and T.-Y.Y. acknowledge support from National Creative Research Initiative Program (Center for Single-Molecule Systems Biology). W.

## Abstract

The quantum yield of a fluorophore is reduced when two or more identical fluorophores are in close proximity to each other. The study of protein folding or particle aggregation is can be done based on this above-mentioned phenomenon—called self-quenching. However, it is challenging to characterize the self-quenching of a fluorophore at high concentrations because of the inner filter effect, which involves depletion of excitation light and re-absorption of emission light. Herein, a novel method to directly evaluate the self-quenching behavior of fluorophores was developed. The evanescent field from an objective-type total internal reflection fluorescence (TIRF) microscope was used to reduce the path length of the excitation and emission light to ~100 nm, thereby supressing the inner filter effect. Fluorescence intensities of sulforhodamine B, fluorescein isothiocyanate (FITC), and calcein solutions with concentrations ranging from 1 μM to 50 mM were directly measured to evaluate the concentration required for 1000-fold degree of self-quenching and to examine the different mechanisms through which the fluorophores undergo self-quenching.

## Introduction

The signal-to-noise ratio (SNR) and the detection sensitivity of fluorescence spectroscopy are significantly higher than those of other detection techniques. Nevertheless, a good SNR is required to identify the fluorescence signal with high-fidelity [1, 2]. A facile method for the improvement of the SNR, and therefore the detection limit, involves increasing the number of fluorophores with high-density labeling. However, it is well-known that because of self-quenching, the intensity of fluorescence emission decreases at higher concentrations of the fluorophore solution [3–5]. Self-quenching can occur via various mechanisms such as collisions between excited fluorophores, the formation of non-fluorescent dimers, and energy transfer to the nonfluorescent dimers [6–11]. Although self-quenching is generally considered

B. is supported by EPSRC grant EP/P02596X/1. C.J is supported by the KIST Institutional Program (Project No. 2E31092) and the National Research Foundation of Korea (NRF) grant funded by the Ministry of Science and ICT of Korea (Project No. 2017R1A3B1023418).

**Competing interests:** NO authors have competing interests.

as a drawback for achieving high-density labeling [12–14], it can be utilized in beneficial ways. Self-quenching has been utilized for the volumetric and leakage assays of lipid structures [15–18], wherein a fluorophore with high concentration is initially encapsulated inside the membrane to facilitate self-quenching. An increase in the volume or permeability of the membrane causes the concentration of fluorophores inside the membrane to decrease. This leads to an increase in the fluorescence signal due to a decrease in self-quenching. Previous studies also reported that conformational changes can be observed by a change in the self-quenching behavior of two fluorophores attached to different spots in a target molecule even at the single-molecule level [15–17]. Self-quenching has also allowed monitoring protein aggregation in real-time [18–20]. Therefore, characterizing the self-quenching behavior of fluorophores will provide valuable reference data for the design of experiments, which involve high concentrations of fluorophores.

Experimental determination of self-quenching is not trivial because it should be performed at millimolar concentrations of the fluorophore, at which the inner filter effect must also be considered (Fig 1A) [21, 22]. The inner filter effect is a combination of two processes: excitation light depletion and re-absorption of emitted photons. For a typical fluorophore with an extinction coefficient of 10,000 $M^{-1}cm^{-1}$, a significant amount of excitation light depletion and re-absorption of emitted photons will occur at concentrations of >10 µM when the path length is 1 cm. Since both effects depend on the path length, the easiest way to prevent the inner filter effect is to decrease the path length of the cuvette. Nevertheless, a path length of 0.01 cm is only 100 times higher than that of 1 cm, implying that the fluorescence measurements can only be accurate at concentrations of <1 mM. Alternatively, the inner filter effect can also be suppressed by employing a nanostructure to encapsulate a high concentration of the fluorophore and thus, reduce the overall concentration. However, previous studies have revealed that the nanostructure itself changes the local environment of the fluorophores, which in turn affects their photophysical properties [23], as shown in a study where self-quenching was released near the metal surface [24]. Therefore, there is a need to develop an experimental

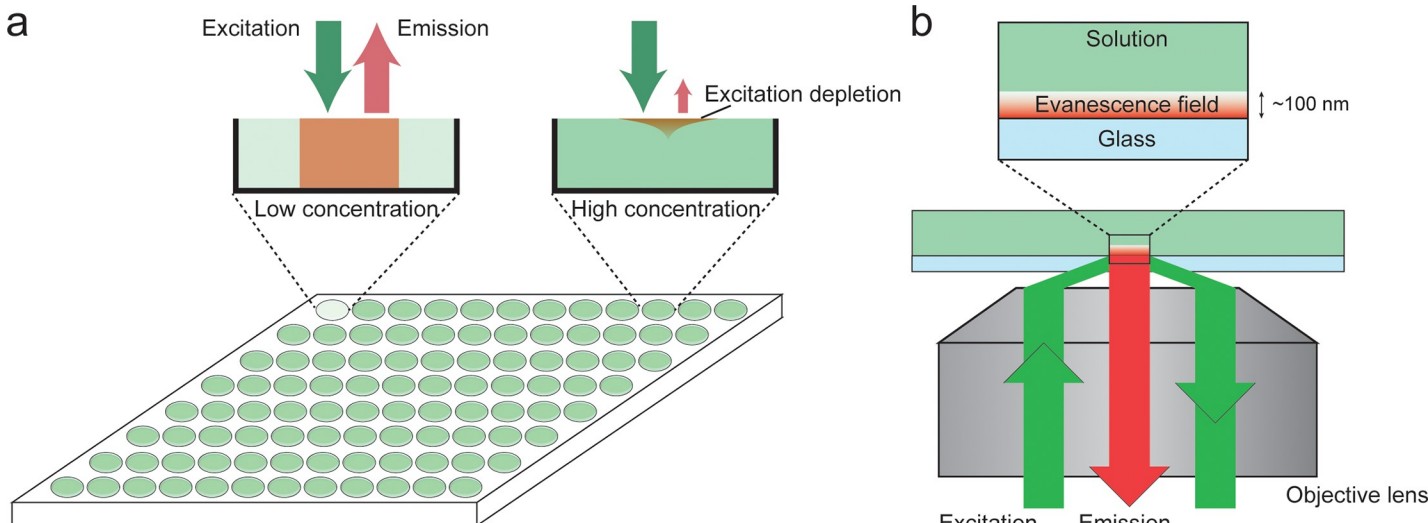

**Fig 1. Comparison between the conventional fluorometric system and the nano-cuvette system.** (a) Fluorescence measurement of solutions with low and high fluorophore concentrations in a microplate reader. Compared with solutions of low concentration in which all the fluorophores are excited (left), at high concentration, only a small portion of fluorophores near-surface are excited owing to the rapid absorption of the excitation light. (b) Schematic representation of the nano-cuvette system. The excitation beam illuminates the chamber in total internal reflection mode which generates an evanescence field with a penetration depth of ~100 nm. With this depth, the emission from all the fluorophores can be collected by the objective lends as no absorption of excitation beam occurs.

method to directly determine the self-quenching of fluorophores in an arbitrary local environment.

In this study, a novel method was developed, wherein the evanescent field from an objective-type total internal reflection fluorescence (TIRF) microscope [25] was used to reduce the effective path length to ~100 nm (Fig 1B). At this path length, the inner filter effect is significantly suppressed and the fluorescence signals of numerous fluorophores can be accurately measured at concentrations up to a 1 M, which is higher than the solubility limit of most fluorophores.

## Materials and methods

### Microplate reader measurement

The fluorescence intensity of solutions with different concentrations of the fluorophores was measured using a microplate reader from PerkinElmer (EnSpire), USA. Fluorophores were dissolved in a buffer solution, which contained 50 mM HEPES (H3375), 100 mM NaCl (S7653), and 5% glycerol (G5516), at pH 7.4. All chemicals used for preparing the buffer solution were purchased from Sigma-Aldrich. A total of 100 μL of each solution was loaded into a 96-well plate (655076 from Greiner) and the fluorescence signals were measured at the following wavelengths with an integration time of 0.1 seconds: excitation 565 nm, emission 586 nm for sulforhodamine B (S1307 from ThermoFisher), excitation 498 nm, emission 517 nm for FITC (F6377 from Sigma-Aldrich), and excitation 493 nm, emission 515 nm for calcein (C0003 from TCI, Japan). All fluorescence signals were corrected with a background signal using the buffer-only solution. The signal was fitted using Origin 8.0 (OriginLab)

### TIRF microscopy

The flow chamber was passivated with polyethylene glycol to minimize the possible interactions between the fluorophores and the glass surface by following a previously described method [26]. 30 μL of each fluorophore solution, prepared with the same buffer as the microplate reader experiment, was injected into a custom-built flow chamber [26]. Fluorescence measurements for FITC were performed by using an objective-type TIR microscope based on Ti-E (Nikon, Japan) with CFI APO TIRF 60XH objective lens while Sulforhodamine b and Calcein were measured using a home-built objective-type TIR microscope based on Olympus IX71 (Japan). The detailed configuration of this microscope can be found elsewhere [26]. In brief, a laser beam was briefly allowed to enter through the backport of the microscope and beam splitters were used to separate the emission wavelength from the laser (Semrock, Di03-R473-t1-25 × 36 for 473 nm and Di02-R532-25 × 36 for 532 nm laser). Fluorescence from the blue (FITC and calcein, 473 nm, Thorlabs, USA) or green (sulforhodamine B, 532 nm, Thorlabs USA) laser excitation was recorded with an electron-multiplying charge-coupled device (EM-CCD, iXon 897+, Andor, Northern Ireland) after using long-pass filters to suppress the excitation laser (Semrock, LP02-473RS-25 for FITC and calcein, LP03-532RS-25 for sulforhodamine B). To minimize the noise, the fluorescence signal from a large number of pixels (262,144 for FTIC and 160,000 for Sulforhodamine b and Calcein) were taken and averaged for each image. At least 3 images were taken for each condition.

## Results

The proposed method can be easily performed by injecting fluorophore solutions of different concentrations into a microfluidic channel and then, recording their fluorescence by using an objective-type TIR microscope (Fig 1B). These measurements were compared with the

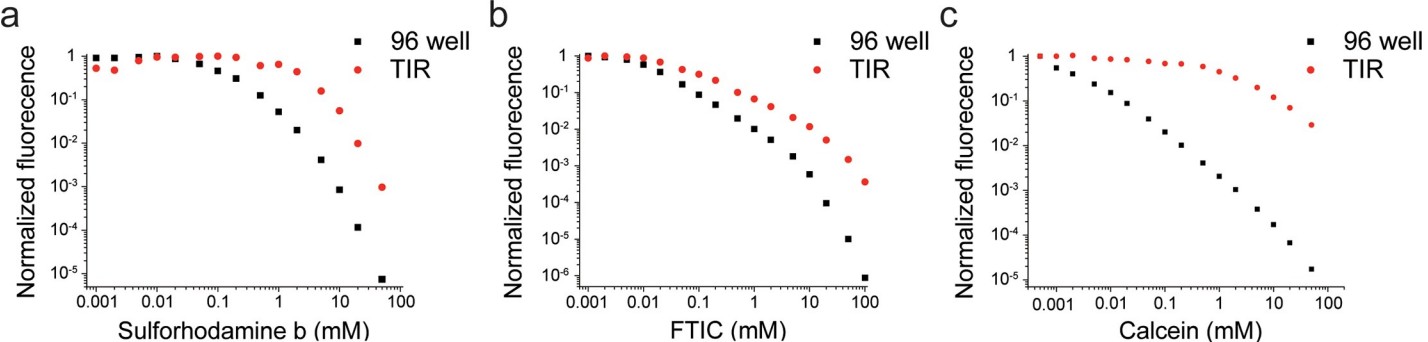

**Fig 2. Normalized fluorescence signal per fluorophore at different concentrations recorded with conventional fluorometer and TIR microscope.** Fluorescence signals from the conventional fluorometer are significantly underestimated by approximately two to three orders of magnitude (black dots). In TIR, 1000-fold self-quenching is observed at 50 mM sulforhodamine B solution (a) and 100 mM fluorescein isothiocyanate (FITC) solution (b). Calcein shows only 100-fold quenching at 50 mM concentration (c). Raw data can be found in S1 Table in S1 File.

fluorescence intensities per fluorophore recorded by a conventional 96-well microplate reader (Fig 2). The fluorescence of three popular fluorophores—sulforhodamine B, FITC, and calcein—was evaluated at concentrations ranging from 1 μM to 50 mM. These fluorophores are water-soluble and self-quenched at high concentrations, making them good agents for bulk self-quenching assays such as volumetric assays.

As expected, the inner filter effect causes the fluorescence signals from the microplate reader (Fig 2, black dots) to be approximately two to three times lower than those recorded by the TIR microscope (Fig 2, red dots). From our TIR measurement, the maximum degree of quenching was 100-fold for 50 mM calcein and 1000-fold for 50 mM sulforhodamine B and 100 mM fluorescein (Fig 2, red dots). While all fluorophores underwent >95% self-quenching at 50 mM, their quenching behavior was not the same. sulforhodamine B and FITC displayed self-quenching at concentrations of >0.5 mM and were 99.9% quenched at 100 mM (Fig 2A and 2B). In contrast, calcein showed rapid self-quenching behavior at 10 μM concentration, but its quantum yield decreased slowly and it was only 99% quenched at 50 mM (Fig 2C).

The degree of quenching was plotted against concentration to examine the self-quenching mechanism of the fluorophores (Fig 3). Notably, the quenching of calcein was linear on concentration, while the quenching of sulforhodamine B and FITC showed higher-order dependency on concentration, suggesting the presence of additional quenching mechanisms (Fig 3,

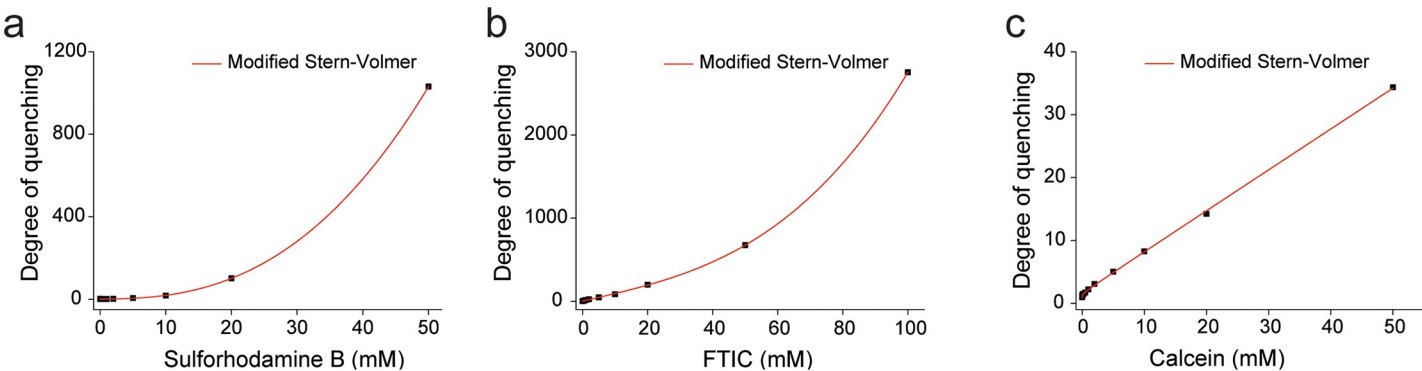

**Fig 3. Degree of quenching against concentration.** All of the curves were fitted with the modified Stern-Volmer equation for self-quenching: $\frac{I_0}{I} \cong c + a[\text{Fluorophore}]_0 + b[\text{Fluorophore}]_0^v$. The value of $v$ was 2.6, for Sulforhodamine B, 3.0 for FITC and 0.04 for calcein.

S1 Fig in S1 File, and S2 Table in S1 File). Sulforhodamine B and FITC showed a higher-order concentration dependency (a, b) while calcein mainly showed a first-order concentration dependency (c). Therefore, we fit the curve with a modified Stern-Volmer equation [27] which is given as $\frac{I_0}{I} \cong 1 + a[\text{Fluorophore}]_0 + b[\text{Fluorophore}]_0^{\nu}$. The $\nu$ was 2.6 and 3.0 for sulforhodamine B and FTIC respectively. As the value of $\nu$ is 1.5 for a static quenching, this higher $\nu$ suggests the formation of quenching centers at high concentrations of fluorophores.

## Discussion and conclusions

Using the evanescence wave from an objective-type TIRF microscope, we directly measured the fluorescence and self-quenching behavior of water-soluble fluorophores at high concentrations. The degree of self-quenching against the concentration is closely related to the quenching mechanism of the fluorophore. In the diffusion mediated self-quenching, the degree of quenching depends on the rate of collisions that lead to the loss of fluorescence. Detailed mathematical models for diffusion-mediated quenching behaviours has been developed [28, 29]. Here, we will consider the simplest case where the fluorophores do not have a binding affinity and lose their fluorescence by collisions. The rate of collision is expressed as follows with $\nu$ being the speed of the fluorophore, $\sigma$ being the collisional cross-section and C being the appropriate conversion constant.

$$k_{collision\ loss} = C \times \nu_{Fluorphore} \times \sigma_{Fluorophore} \times [\text{Fluorophore}]$$

The quantum yield in the presence of collisional quenching can be expressed as $\Phi_c = \frac{k_f}{k_f + k_{nf} + k_{collision\ loss}}$ with $k_f$ being the rate of fluorescent emission process and $k_{nf}$ being the rate of nonfluorescent decay process in the absence of quenching. By comparing this with the quantum yield in the absence of collisional quenching $\Phi_0 = \frac{k_f}{k_f + k_{nf}}$, we get the degree of quenching given as

$$\frac{\Phi_0}{\Phi_C} = \frac{k_f + k_{nf} + k_{collision\ loss}}{k_f + k_{nf}} = 1 + k_{collision\ loss}\frac{1}{k_f + k_{nf}} = 1 + \frac{k_{c_0}}{k_f + k_{nf}}[\text{Fluorophore}] \qquad (1)$$

Therefore, the degree of quenching from the collisional quenching increases linearly with the concentration of fluorophores. This can explain the self-quenching of calcein but not the sulforhodamine B and FTIC. Therefore, we considered additional self-quenching mechanisms such as the formation of non-fluorescent dimers. The formation of dimers, can be described by two equations. First, the formation of the dimers decreases the concentration of the free fluorophores.

$$[\text{Fluorophore}] = [\text{Fluorophore}]_0 - 2[\text{Dimer}]$$

The concentration of dimers is determined by the dissociation constant $K_D$.

$$K_D[\text{Dimer}] = [\text{Fluorophore}][\text{Fluorophore}]$$

After solving the equation (S1 Note in S1 File), we get the degree of quenching due to the formation of non-fluorescent dimers as

$$\frac{[\text{Fluorophore}]_0}{[\text{Fluorophore}]} = \frac{1}{\sqrt{\left(\frac{K_D}{[\text{Fluorophore}]_0}\right)^2 + 8\frac{K_D}{[\text{Fluorophore}]_0}} - \frac{K_D}{[\text{Fluorophore}]_0}} \qquad (2)$$

As we were not able to fit the data from sulforhodamine B and FTIC using this function (S2 Fig in S1 File), this suggests the formation of nonfluorescent multimers at high concentrations and potentially higher order multimers for FTIC.

More information on the mechanism of self-quenching can be obtained by using fluorescence techniques such as fluorescence lifetime and fluorescence anisotropy measurement. For example, if the main mechanism of self-quenching is collisional, the anisotropy measurement will not change but we should see a decreased lifetime in the fluorescence lifetime measurement. Or if the formation of stable dye aggregate is the main reason for self-quenching, we will see multiple lifetimes in the fluorescence lifetime measurement while the anisotropy could potentially change if the aggregate is fluorescent.

Simultaneous measurement of concentrations could not be carried out because of the dynamic pixel range of the camera. The EM-CCD camera gave a 14-bit integer as the output data and the maximum pixel value was 16384. Considering the dark shot noise, the effective dynamic range for one pixel is $\sim 10^3$. Modulating the intensity of the excitement laser and the integration time allowed the evaluation over a wider range of concentrations. Also, the noise was detected at low concentrations because of the lack of fluorophores inside the nano-cuvette. The approximate excitation volume is:

$$400 \text{ pixels } (160 \text{ nm}) \times 400 \text{ pixels } (160 \text{ nm}) \times 100 \ nm = 4 \times 10^{-13} l.$$

Assuming that a good SNR can be obtained if there is one fluorophore per 10 pixels, then 16,000 fluorophores are required for the calculated volume. This corresponds to a low concentration, i.e., 0.27 μM, at which self-quenching does not occur. Compared with other previously reported methods, such as liposome encapsulation [10, 30] and theoretical correction [31], the method proposed herein is a simpler and more direct experimental approach to measure the self-quenching of soluble fluorophores over a wide range of concentrations, in an arbitrary environment. Since the proposed method is based on an objective-type TIRF microscope, more information can be retrieved with the help of spectroscopic detectors and polarization measurements, to provide insight into the photophysical processes that occur during the self-quenching process.

## Supporting information

**S1 File.**
(DOCX)

## Author Contributions

**Conceptualization:** Wooli Bae.

**Data curation:** Wooli Bae, Cherlhyun Jeong.

**Funding acquisition:** Tae-Young Yoon, Cherlhyun Jeong.

**Investigation:** Wooli Bae.

**Methodology:** Wooli Bae, Cherlhyun Jeong.

**Project administration:** Wooli Bae.

**Supervision:** Tae-Young Yoon.

**Validation:** Cherlhyun Jeong.

**Writing – original draft:** Wooli Bae.

**Writing – review & editing:** Wooli Bae, Tae-Young Yoon, Cherlhyun Jeong.

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
