## [Decision Letter · Decision Letter 0]

24 Nov 2020

PONE-D-20-29041

Direct evaluation of self-quenching behavior of fluorophores at high concentrations using an evanescent field <o:p></o:p>

PLOS ONE

Dear Dr. Bae,

Thank you for submitting your manuscript to PLOS ONE. After careful consideration, we feel that it has merit but does not fully meet PLOS ONE’s publication criteria as it currently stands. Therefore, we invite you to submit a revised version of the manuscript that addresses the points raised during the review process.

We look forward to receiving your revised manuscript.

Kind regards,

Xiaowei Zhang, Ph.D.

Academic Editor

PLOS ONE

Journal Requirements:

2. Please amend either the title on the online submission form (via Edit Submission) or the title in the manuscript so that they are identical.

Reviewers' comments:

Reviewer's Responses to Questions

**Comments to the Author**

1. Is the manuscript technically sound, and do the data support the conclusions?

Reviewer #1: Partly

Reviewer #2: Partly

2. Has the statistical analysis been performed appropriately and rigorously? 

Reviewer #1: Yes

Reviewer #2: Yes

3. Have the authors made all data underlying the findings in their manuscript fully available?

Reviewer #1: Yes

Reviewer #2: Yes

4. Is the manuscript presented in an intelligible fashion and written in standard English?

Reviewer #1: Yes

Reviewer #2: Yes

5. Review Comments to the Author

Reviewer #1: Comments to the authors:

This paper demonstrated a novel method to directly evaluate the self-quenching behavior of soluble fluorophores. Compared to other methods, such as liposome encapsulation or theoretical correction, this method is simpler and direct experimental way to evaluate the self-quenching of fluorophores at arbitrary environment. The actuality of the research topic and the quality of the present study matches the journal’s profile and audience. However, the following comments should be properly addressed before the paper can be considered for publication in Plos One.

1. The comparison between the conventional fluorometric system and the nano-cuvette system is not a perfect illustration in Figure 1. The principle and annotation of the TIRF microscopy should be added and discussed.

2. According to what I know, various mechanisms will lead to the self-quenching effect, especially in the high-density fluorophores. In Line 125, the authors mentioned that the plots of sulforhodamine B and FITC exhibited their second-order dependency on concentration. Why do the second-order dependency on concentration indicate the self-quenching occurred via the resonance energy transfer mechanism, but not the collision mechanism between excited fluorophores or the formation of nonfluorescent dimers? Clarify it and add related explanations on the self-quenching reason. Here the authors should spend more time showcasing how the test result determinate the different self-quenching mechanisms.

3. Where is the Conclusion Part for this paper? Add the conclusion for this paper.

4. The depth of the discussion is not strong enough. One would expect some modelling to explain the different self-quenching mechanisms.

Reviewer #2: I have read the manuscript. The idea about employing total internal reflection fluorescence (TIRF) microscope to directly evaluate self-quenching of fluorophores at large range of concentration is interesting. However, some issues cannot be ignored.

1. I see that the measured emission wavelength between microplate reader measurement and TIRF microscopy are different. Why not detect the same emission wavelength? Does the measured emission wavelength of three kinds of fluorophores located at characteristic emission wavelength?

2. The verification of the results measured from TIRF microscopy is lack. Kindly do some lifetime studies with different concentrations.

3. Self-quenching research always has been an interesting topic. The introduction of progress in recent years is necessary.

4. There are some grammar errors in the manuscript. Please update.

6. PLOS authors have the option to publish the peer review history of their article (what does this mean?). If published, this will include your full peer review and any attached files.

Reviewer #1: No

Reviewer #2: No

---

## [Author Response · Author response to Decision Letter 0]

3 Feb 2021

Reviewer 1

This paper demonstrated a novel method to directly evaluate the self-quenching behavior of soluble fluorophores. Compared to other methods, such as liposome encapsulation or theoretical correction, this method is simpler and direct experimental way to evaluate the self-quenching of fluorophores at arbitrary environment. The actuality of the research topic and the quality of the present study matches the journal’s profile and audience. However, the following comments should be properly addressed before the paper can be considered for publication in Plos One.

1. The comparison between the conventional fluorometric system and the nano-cuvette system is not a perfect illustration in Figure 1. The principle and annotation of the TIRF microscopy should be added and discussed.

Response : We have updated the Figure 1 to have more detailed description on the TIRF microscopy. 

2. According to what I know, various mechanisms will lead to the self-quenching effect, especially in the high-density fluorophores. In Line 125, the authors mentioned that the plots of sulforhodamine B and FITC exhibited their second-order dependency on concentration. Why do the second-order dependency on concentration indicate the self-quenching occurred via the resonance energy transfer mechanism, but not the collision mechanism between excited fluorophores or the formation of nonfluorescent dimers? Clarify it and add related explanations on the self-quenching reason. Here the authors should spend more time showcasing how the test result determinate the different self-quenching mechanisms.

Response : As requested, we have added a section to explain various mechanisms and their effect on the self-quenching curve. However, some of the mechanisms do not have a simple analytical solution. Therefore, we have fit the data with an extended Stern-Volmer equation - I_0/I≅1+a[Fluorophore]_0+b[Fluorophore]_0^ν. This not only gave us better fits but also clearly demonstrated presence of multimers in sulforhodamine B and FTIC. 

“The degree of quenching was plotted against concentration to examine the self-quenching mechanism of the fluorophores (Figure 3). Notably, the quenching of calcein was linear on concentration, while the quenching of sulforhodamine B and FITC showed higher-order dependency on concentration, suggesting the presence of additional quenching mechanisms (Fig 3, Supplementary Fig S1, and Supplementary Table 2). Sulforhodamine B and FITC showed a higher-order concentration dependency (a, b) while calcein mainly showed a first-order concentration dependency (c). Therefore, we fit the curve with a modified Stern-Volmer equation[27] which is given as I_0/I≅1+a[Fluorophore]_0+b[Fluorophore]_0^ν. The ν was 2.6 and 3.0 for sulforhodamine B and FTIC respectively. As the value of ν is 1.5 for a static quenching, this higher ν suggests the formation of quenching centers at high concentrations of fluorophores. “

“Using the evanescence wave from an objective-type TIRF microscope, we directly measured the fluorescence and self-quenching behavior of water-soluble fluorophores at high concentrations. The degree of self-quenching against the concentration is closely related to the quenching mechanism of the fluorophore. In the diffusion mediated self-quenching, the degree of quenching depends on the rate of collisions that lead to the loss of fluorescence. Detailed mathematical models for diffusion-mediated quenching behaviours has been developed [28, 29]. Here, we will consider the simplest case where the fluorophores do not have a binding affinity and lose their fluorescence by collisions. The rate of collision is expressed as follows with v being the speed of the fluorophore, σ being the collisional cross-section and C being the appropriate conversion constant. 

k_(collision loss)=C×v_Fluorphore×σ_Fluorophore×[Fluorophore]

The quantum yield in the presence of collisional quenching can be expressed as Φ_c=k_f/(k_f+k_nf+k_(collision loss) ) with k_f being the rate of fluorescent emission process and k_nf being the rate of nonfluorescent decay process in the absence of quenching. By comparing this with the quantum yield in the absence of collisional quenching Φ_0=k_f/(k_f+k_nf ), we get the degree of quenching given as 

Φ_0/Φ_C =(k_f+k_nf+k_(collision loss))/(k_f+k_nf ) 〖=1+k〗_(collision loss) 1/(k_f+k_nf )=1+k_(c_0 )/(k_f+k_nf ) [Fluorophore] (1)

Therefore, the degree of quenching from the collisional quenching increases linearly with the concentration of fluorophores. This can explain the self-quenching of calcein but not the sulforhodamine B and FTIC. Therefore, we considered additional self-quenching mechanisms such as the formation of non-fluorescent dimers. The formation of dimers, can be described by two equations. First, the formation of the dimers decreases the concentration of the free fluorophores. 

[Fluorophore]= [Fluorophore]_0-2[Dimer]

The concentration of dimers is determined by the dissociation constant K_D.

K_D [Dimer]=[Fluorophore][Fluorophore]

After solving the equation (Supplementary note 1), we get the degree of quenching due to the formation of non-fluorescent dimers as

[Fluorophore]_0/([Fluorophore])=1/(√(((K_D^ )/[Fluorophore]_0 )^2+8 (K_D^ )/[Fluorophore]_0 )-(K_D^ )/[Fluorophore]_0 ) (2)

As we were not able to fit the data from sulforhodamine B and FTIC using this function (Supplementary Fig. 2), this suggests the formation of nonfluorescent multimers at high concentrations and potentially higher order multimers for FTIC. 

More information on the mechanism of self-quenching can be obtained by using fluorescence techniques such as fluorescence lifetime and fluorescence anisotropy measurement. For example, if the main mechanism of self-quenching is collisional, the anisotropy measurement will not change but we should see a decreased lifetime in the fluorescence lifetime measurement. Or if the formation of stable dye aggregate is the main reason for self-quenching, we will see multiple lifetimes in the fluorescence lifetime measurement while the anisotropy could potentially change if the aggregate is fluorescent.“

3. Where is the Conclusion Part for this paper? Add the conclusion for this paper.

Response : We added a conclusion and discussion part as requested. 

4. The depth of the discussion is not strong enough. One would expect some modelling to explain the different self-quenching mechanisms.

Response : We agree with the reviewer’s comment regarding on the depth of the discussion part. We have added analytical solutions on different self-quenching mechanisms and additional references in the discussion part.

Reviewer 2

I have read the manuscript. The idea about employing total internal reflection fluorescence (TIRF) microscope to directly evaluate self-quenching of fluorophores at large range of concentration is interesting. However, some issues cannot be ignored.

1. I see that the measured emission wavelength between microplate reader measurement and TIRF microscopy are different. Why not detect the same emission wavelength? Does the measured emission wavelength of three kinds of fluorophores located at characteristic emission wavelength?

Response : Tor the TIRF microscopy, we have chosen the best set of laser and emission filter from our list. This wavelength is also typically used by other experiments involving a laser and microscope. In comparison, we have more freedom in the wavelength for the microplate reader-based experiments as it is equipped with a monochromator. Therefore, the wavelength was determined by considering the spectrum of fluorophores and a guideline from the manufacturer requiring certain amount of separation between the excitation and emission wavelength. 

2. The verification of the results measured from TIRF microscopy is lack. Kindly do some lifetime studies with different concentrations.

Response : As the reviewer said, the fluorescence lifetime measurement would confirm the occurrence of self-quenching as well as determining its mechanism. However, due to the current Covid-related situation in UK, we were not able to reach out to perform a lifetime measurement. We have added a section discussing on the benefit of lifetime measurement in the discussion. 

“More information on the mechanism of self-quenching can be obtained by using fluorescence techniques such as fluorescence lifetime and fluorescence anisotropy measurement. For example, if the main mechanism of self-quenching is collisional, the anisotropy measurement will not change but we should see a decreased lifetime in the fluorescence lifetime measurement. Or if the formation of stable dye aggregate is the main reason for self-quenching, we will see multiple lifetimes in the fluorescence lifetime measurement while the anisotropy could potentially change if the aggregate is fluorescent.“

3. Self-quenching research always has been an interesting topic. The introduction of progress in recent years is necessary.

Response : As requested, we have cited more recent works on self-quenching in the introduction and discussion part. 

4. There are some grammar errors in the manuscript. Please update.

Response : We have gone through the manuscript and corrected as much grammar errors as possible.

---

## [Editor Report · Decision Letter 1]

5 Feb 2021

Direct evaluation of self-quenching behavior of fluorophores at high concentrations using an evanescent field

PONE-D-20-29041R1

Dear Dr. Bae,

We’re pleased to inform you that your manuscript has been judged scientifically suitable for publication and will be formally accepted for publication once it meets all outstanding technical requirements.

Kind regards,

Xiaowei Zhang, Ph.D.

Academic Editor

PLOS ONE

Additional Editor Comments (optional):

The author's responses to the reviewer's queries are satisfactory. Now the revised improved well with the better scientific discussions.

---

## [Editor Report · Acceptance letter]

10 Feb 2021

PONE-D-20-29041R1 

Direct evaluation of self-quenching behavior of fluorophores at high concentrations using an evanescent field 

Dear Dr. Bae:

I'm pleased to inform you that your manuscript has been deemed suitable for publication in PLOS ONE. Congratulations! Your manuscript is now with our production department. 

Kind regards, 

on behalf of

Dr. Xiaowei Zhang 

Academic Editor

PLOS ONE